# Iron Oxide/Chitosan Magnetic Nanocomposite Immobilized Manganese Peroxidase for Decolorization of Textile Wastewater

**Saifeldin M. Siddeeg** [1,2], **Mohamed A. Tahoon** [1], **Wissem Mnif** [3,4] **and Faouzi Ben Rebah** [1,5,*]

1   Department of Chemistry, College of Science, King Khalid University, P.O. Box 9004, Abha 61413, Saudi Arabia; saif.siddeeg@gmail.com (S.M.S.); tahooon_87@yahoo.com (M.A.T.)
2   Chemistry and Nuclear Physics Institute, Atomic Energy Commission, P.O. Box 3001, Khartoum 11111, Sudan
3   Department of Chemistry, Faculty of Sciences and Arts in Balgarn, University of Bisha, P.O. BOX 199, Bisha 61922, Saudi Arabia; w_mnif@yahoo.fr or wmoneef@ub.edu.sa
4   LR11-ES31 Laboratory of Biotechnology and Valorisation of Bio-Geo Resources, Higher Institute of Biotechnology of Sidi Thabet, BiotechPole of Sidi Thabet, University of Manouba, Biotechpole Sidi Thabet, Ariana 2020, Tunisia
5   Higher Institute of Biotechnology of Sfax (ISBS), Sfax University, P.O. Box 263, Sfax 3000, Tunisia
*   Correspondence: benrebahf@yahoo.fr

**Abstract:** Because of its effectiveness in organic pollutant degradation, manganese peroxidase (MnP) enzyme has attracted significant attention in recent years regarding its use for wastewater treatment. Herein, MnP was extracted from *Anthracophyllum discolor* fungi and immobilized on the surface of magnetic nanocomposite $Fe_3O_4$/chitosan. The prepared nanocomposite offered a high surface area for MnP immobilization. The influence of several environmental factors like temperature, pH, as well as storage duration on the activity of the extracted enzyme has been studied. Fourier transmission infrared spectroscopy (FT-IR), scanning electron microscope (SEM), X-ray diffraction (XRD), and transmission electron microscope (TEM) techniques were used for the characterization of the prepared MnP/$Fe_3O_4$/chitosan nanocomposite. The efficiencies of the prepared MnP/$Fe_3O_4$/chitosan nanocomposite for the elimination of reactive orange 16 (RO 16) and methylene blue (MB) industrial dyes were determined. According to the results, the immobilization of MnP on $Fe_3O_4$/chitosan nanocomposite increases its capacity to decolorize MB and RO 16. This nanocomposite allowed the removal of 96% ± 2% and 98% ± 2% of MB and RO 16, respectively. The reusability of the synthesized nanocomposite was studied for five successive cycles showing the ability to retain its efficiency even after five cycles. Thus, the prepared MnP/$Fe_3O_4$/chitosan nanocomposite has potential to be a promising material for textile wastewater bioremediation.

**Keywords:** bioremediation; nanocomposites; chitosan; enzymes; microorganisms

## 1. Introduction

Urbanization and contamination of surface drinking water resources have led to a crucial problem concerning the provision of safe water supplies in our contemporary world [1]. Uncontrolled discharges of municipal and industrial wastes in the environment can lead to the introduction of high concentrations of potentially toxic pollutants into water resources. Among the most harmful types of wastewater released are textile effluents that contain many dangerous artificial dyes like reactive

orange 16 (RO 16) and methylene blue (MB). Due to its low prices, cationic dye MB is easily acquired and considered to be one of the most popular clothing colorants [2]. The industrial release of MB can disrupt ecosystem balance and damage the environment, due to its carcinogenic effects [3,4]. These effects are accompanied by several symptoms including severe headaches, skin irritation, and acute diarrhea [5]. RO 16 is an anionic azo dye widely used in the dyeing process [6]. The discharge of RO 16 effluent into natural water bodies can cause many problems, such as the reduction of photosynthetic activity by reducing sunlight transmission, negatively affecting the survival of aquatic life. Moreover, azo dyes have been shown to have mutagenic and carcinogenic effects [7,8]. Therefore, dyes must be removed before effluent discharge into the environment, as these molecules are hard to decompose in natural conditions (pH, temperature, etc.) or with conventional methods of treatment [9].

Electrochemical processes, exchanging ions, flotation, coagulation-flocculation, biotic processing, adsorption, reverse osmosis, as well as separation membranes, were widely used for the elimination of organic and inorganic pollutants [10–16]. However, the adsorption technique is the most widely used technique among all methods due to its cheapness, simplicity, and effectiveness [17]. Recently, researchers all over the world focused their efforts to introduce advanced adsorbents for wastewater treatment [18,19]. In this context, enzymes immobilized on solid supports were more widely used than free enzymes, since the immobilized enzyme can be easily separated from the reaction solution using various physical methods, such as centrifugation and filtration [20–22]. Interestingly, nanoparticles have emerged as an efficient tool to generate excellent support for enzymes offering various advantages. Benefits include the ability for a quick biomaterial recuperation using an external magnet, the higher amount of binding enzymes offered by magnetic iron oxide nanoparticles, the low operation price, and the reduction of both fouling and diffusion problems [23–29]. The existence of functional groups on nanoparticle surfaces is vital for their use as supports for bioactive component immobilization. Therefore, the existence of functional groups on the nanoparticle surfaces is intended to increase the potential for loading and stabilization of immobilized biomolecules. Generally, magnetic iron oxide nanoparticles were functionalized using synthetic and natural polymers such as chitosan. Chitosan, a type of hydrocolloid, is a biopolymer of linear positively charged polysaccharide. Highly reactive amino groups are present, increasing its adsorption capacity toward pollutants and allowing it to act as a bio-adsorbent [30]. Nevertheless, its high solubility in acids, tendency to undergo clumping, weak mechanical properties, and difficult separation, limit its use as an adsorbent alone. Chitosan is often used combined with other materials [31] and different forms of chitosan-based material gels, flakes, and powders were applied for the immobilization of many enzymes [32,33]. In recent years, manganese peroxidase (MnP) has attracted the attention of researchers due to its ability to remove various organic contaminants including anthracene [34], phenanthrene [35] as well as polyaromatic hydrocarbons [36]. However, the use of MnP in the industrial applications faces some difficulties such as the low operational stability and the high cost, which can be overcome by the immobilization of the enzyme on the surface of nanocomposites [37].

In the present study we examined the efficiency of MnP enzyme extracted from the white root fungus of *Anthracophyllum discolor* toward the elimination of organic dyes RO 16 and MB from textile wastewater. The performance of the immobilized enzyme on the surface of magnetic nanocomposite $Fe_3O_4$/chitosan was studied and compared to the free enzyme.

## 2. Materials and Methods

### 2.1. Chemicals and Reagents

Ferric chloride ($FeCl_3$) was provided through Xilong Scientific Co. (Shantou, China), while ferrous sulfate ($FeSO_4$), glutaraldehyde, and mineral oil were supplied from Sigma–Aldrich (Missouri, USA). The polymer chitosan, extracted from crab shells, was supplied from Winlab Company (Leicester, England). Acetone solvent, hydrogen peroxide, aqueous ammonia solution (33%) and manganese sulfate ($MnSO_4$) were supplied via Al-Nasr Co., Helwan, Egypt. Methylene blue (MB), a cationic

thiazine dye, and N-(3-Dimethylaminopropyl)-N′-ethylcarbodiimide hydrochloride were supplied via Merck (Hohenbrunn, Germany). Reactive orange 16 (RO 16), an anionic single azo dye, and Tween 80 were supplied from ACROS, Organics (Morris Plains, NJ, USA). All the used materials were of analytical grade and experimentally consumed as supplied without additional purification.

### 2.2. MnP Extraction from Anthracophyllum Discolor

MnP enzyme was extracted from the white root fungus *A. discolor* strain obtained from the microbiology laboratory (Biology Department, Mansoura University). The fungus was allowed to grow at 5 °C inside tubes in the presence of media containing 10 g/L glucose (Merck, Hohenbrunn, Germany), 15 g/L agar (Merck, Hohenbrunn, Germany) and 30 g/L malt extract. The strain culture was then transferred to Petri dishes (Shanghai Joylab Medical Instruments Co., Ltd, Shanghai, China) and incubated for six days at 30 °C. Subsequently, a Kirk media containing 0.06% (*v/v*) surfactant tween 80, 255 μM of manganese sulfate (Al-Nasr Co., Helwan, Egypt) as MnP enzyme production inducer, and 33 g of wheat grains was prepared, and 95 mL of the Kirk media was inoculated by the mycelium of *A. discolor* and incubated at 30 °C for 11 days. The partially purified MnP enzyme was obtained from the media after growth. The growth media was ultra-purified using three successive filters; 1.2 μm Whatman glass filter (GE Healthcare Bio-Sciences, Pittsburgh, USA), an Amicon membrane, and 0.2 μm mixed ester-cellulose filter (Merck, Hohenbrunn, Germany). The ultra-purified enzyme was tested for MnP activity via MBTH-DMAB (3-methyl-2-benzothiazolinone hydrazine and 3-(dimethylamino)benzoic acid) (Sigma Aldrich, Steinheim, Germany) method [38], in which 10 μL hydrogen peroxide (10 mM) (Al-Nasr Co., Helwan, Egypt) was added to a mixture containing $MnSO_4$ (20 mM, 30 μL) (Al-Nasr Co., Helwan, Egypt), ultra-purified filtrate (100 μL) (extracted from the white root fungus *A. discolor* ), MBTH (1.5 mM, 100 μL) (Sigma Aldrich, Steinheim, Germany), and DMAB (6.6 mM, 300 μL, buffer of succinate-lactate (100 mM, pH = 4.5, 1460 μL). A purple color was produced and measured by absorbance. Thus, the enzymatic activity was determined by reading absorbance at $\lambda = 590$ nm and was calculated with the extinction coefficient ($\varepsilon = 53,000$ $M^{-1}$ $cm^{-1}$) according to the literature. The amount of enzyme needed for one μM of the reaction product in one minute is equal to one unit [38].

### 2.3. Preparation of Fe₃O₄/Chitosan Magnetic Nanocomposite

Co-precipitation technique was used for $Fe_3O_4$ nanoparticle preparation. $FeSO_4$ and $FeCl_3$ solutions were mixed (1:2 molar ratios) and supplemented by aqueous ammonia with continuous stirring under nitrogen (Smgases, Beni Suef, Egypt) atmosphere for 30 min. The resulted nanoparticles were recuperated using a permanent magnet (Risheng Magnets International Co.,Ltd, Ningbo, China) and washed several times with distilled water. The $Fe_3O_4$/chitosan nanocomposite was prepared via reversed-phase suspension method in which 200 mg of washed iron oxide nanoparticles were mixed with 55 mL of mineral oil (Sigma–Aldrich, Missouri, USA) containing 0.6 mL of tween 80 (Acros Organics, Morris Plains, NJ, USA). After that, 1% (*w/v*) of chitosan (Winlab Company, Leicester, England) was added to the mixture. The synthesized $Fe_3O_4$/chitosan nanocomposite was sonicated and stirred for 35 min. The mixture was supplemented with 3.5 mL of 25% (*w/v*) glutaraldehyde solution (Sigma–Aldrich, St. Louis, MO, USA) and stirred for 5 h. Finally, the synthesized $Fe_3O_4$/chitosan nanocomposite was isolated using a magnet, washed many times using acetone, and dried at 45 °C in a vacuum. The produced $Fe_3O_4$ nanoparticles and $Fe_3O_4$/chitosan nanocomposite were characterized using different techniques including Fourier transform infrared (FT-IR) spectroscopy (Bruke EQUINOX 55, Karlsruhe, Germany), transmission electron microscope (TEM, Technai $G^2$ F30, Hillsboro, USA), scanning electron microscope (SEM, LEO 1450 VP, England), as well as X-ray diffraction (XRD, Rigaku MiniFlex 600 X-ray, USA).

### 2.4. Immobilization of MnP on the Fe₃O₄/Chitosan Nanocomposite

2 mL of phosphate buffer (Al-Nasr Co., Helwan, Egypt) (0.04 M, pH = 6) was used to disperse 50 mg of nanocomposite in an ultrasonic bath at 4 °C to immobilize MnP on the Fe₃O₄ nanocomposite. Then, 0.6 mL of 2.5% N-(3-dimethylaminopropyl)-N′-ethylcarbodiimide hydrochloride (Merck, Hohenbrunn, Germany) in 2 mL phosphate buffer was added to the previous solution and ultra-sonicated for a half-hour. The obtained mixture was stirred for 7 h at 1000 rpm and 4 °C. After that, the nanocomposite was collected with a permanent magnet (Risheng Magnets International Co.,Ltd, Ningbo, China) and washed several times in the phosphate buffer. For each 10 mg of the nanocomposite, 250 μL of the filtrate was added and stirred at 200 rpm and 30 °C for 30 min. Subsequently, the nanocomposite was collected by the magnet and washed by buffer solution (pH = 4.5) of sodium acetate (Al-Nasr Co., Helwan, Egypt) (50 mM) until the activity of the enzyme disappeared from the resulted solution. Finally, the obtained MnP/Fe₃O₄/chitosan nanocomposite was dried at 35 °C. The effect of various operating conditions including pH (3.5–9.5), temperature (30–70 °C) for 1 h, and storage duration at 20 °C (for 14 days) on immobilized MnP as well as free MnP were studied. Each experiment was performed in triplicate.

### 2.5. Wastewater Treatment Experiment

In order to study the removal efficiency of MnP, Fe₃O₄/chitosan nanocomposite, and MnP/Fe₃O₄/chitosan nanocomposite toward dyes RO 16 and MB, synthetic wastewater containing 50 mg/L of each dye was prepared. A 100 mL Erlenmeyer flask (Sigma Aldrich, Steinheim, Germany) was used in the adsorption experiment, in which 50 mL of synthetic wastewater was mixed with 10 mg of Fe₃O₄/chitosan nanocomposite, and MnP/Fe₃O₄/chitosan nanocomposite. In the case of the free MnP, the same volume as the immobilized ultra-purified filtrate was used. The wastewater decolorization experiment was conducted at pH 7, temperature 27 °C, 100 rpm and, for a contact time ranged from 10 to 50 min. The dye concentrations before and after adsorption were analyzed using a UV/VIS spectrophotometer (SHIMADZU, UV-2401pc, Addison, IL, USA) at 495 nm and 660 nm for RO 16 and MB, respectively. After each experiment, the magnetic nanocomposites were collected using a permanent magnet. Each experiment was performed in triplicate.

## 3. Results and Discussion

### 3.1. Characterization of the Synthesized Nanocomposite

Different techniques for characterization including XRD, SEM, FT-IR, and TEM were used to identify chitosan, synthesized Fe₃O₄ nanoparticles, and synthesized Fe₃O₄/chitosan nanocomposite. SEM images of Fe₃O₄ and Fe₃O₄/chitosan nanoparticles are presented in Figure 1a,b, respectively. SEM showed a homogeneous distribution of Fe₃O₄ nanoparticles, which is related to the presence of chitosan offering high surface for the immobilization of the MnP enzyme. Furthermore, a spherical shape of iron oxide nanoparticles and a size less than 35 nm were determined from the SEM image. Based on the TEM images (Figure 1c,d for Fe₃O₄ and Fe₃O₄/chitosan, respectively), the prepared nanoparticles showed uniform spherical morphology with crystalline structure.

FT-IR Bands for Fe₃O₄, chitosan and Fe₃O₄/chitosan nanocomposite are presented in Figure 2a. According to the FT-IR of the magnetic Fe₃O₄ nanoparticles, the 465 cm$^{-1}$ band beside the 579 cm$^{-1}$ band denotes octahedral and tetrahedral sites Fe–O bands [39]. Several bands were noticed representing the different functional groups of chitosan and Fe₃O₄/chitosan nanocomposite. The bands include stretching and bending C=O vibrations with 1530 cm$^{-1}$ and 1626 cm$^{-1}$, wide amino and O–H vibrations with an interval of 3000 cm$^{-1}$ –3620 cm$^{-1}$ and vibrational C–O band with 1069 cm$^{-1}$ [40]. Additionally, the presence of Fe–O bands in the IR spectrum of Fe₃O₄/chitosan nanocomposite indicates the in-situ structure of Fe₃O₄ nanoparticles inside Fe₃O₄/chitosan nanocomposite.

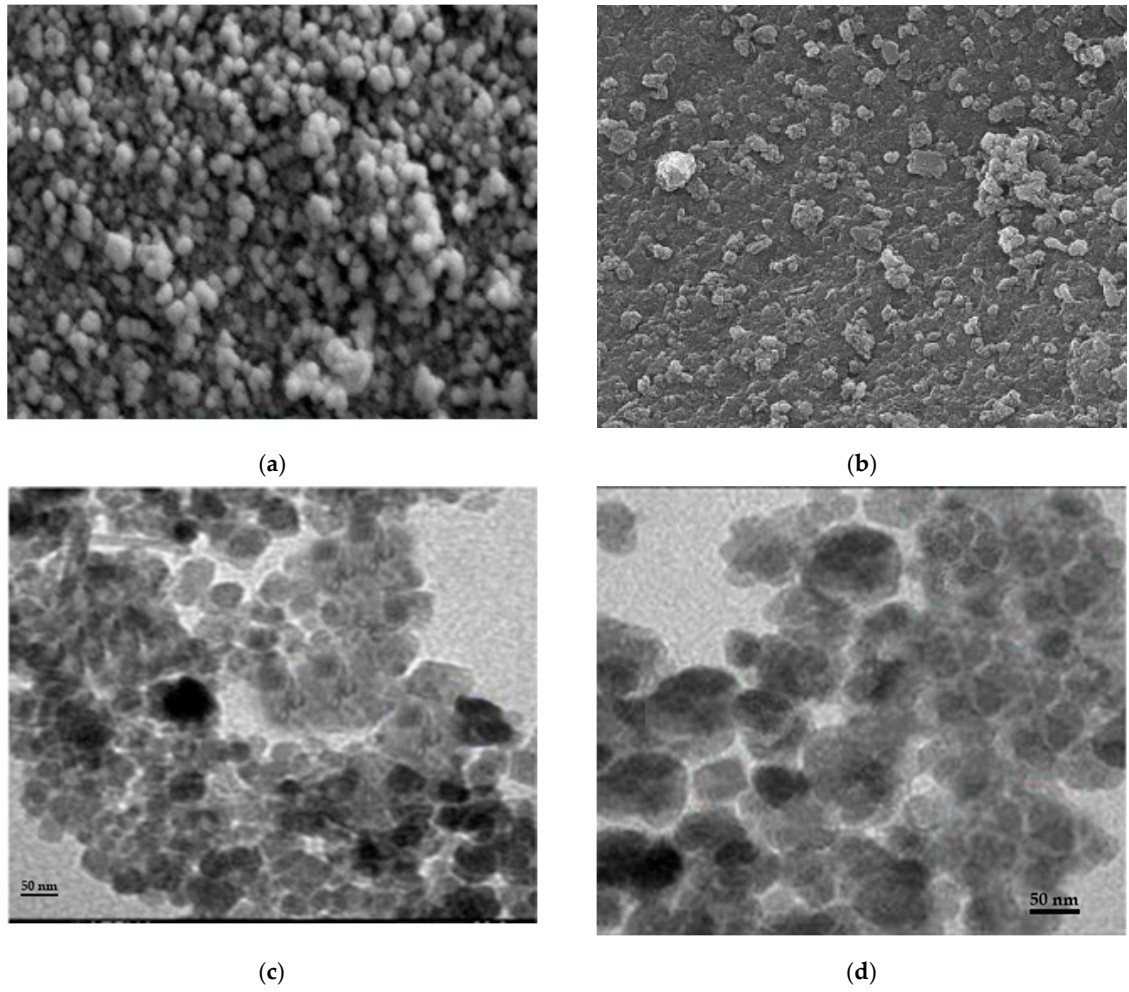

**Figure 1.** (**a**) SEM image of Fe₃O₄; (**b**) SEM image of Fe₃O₄/chitosan; (**c**) TEM image of Fe₃O₄; (**d**) TEM image of Fe₃O₄/chitosan.

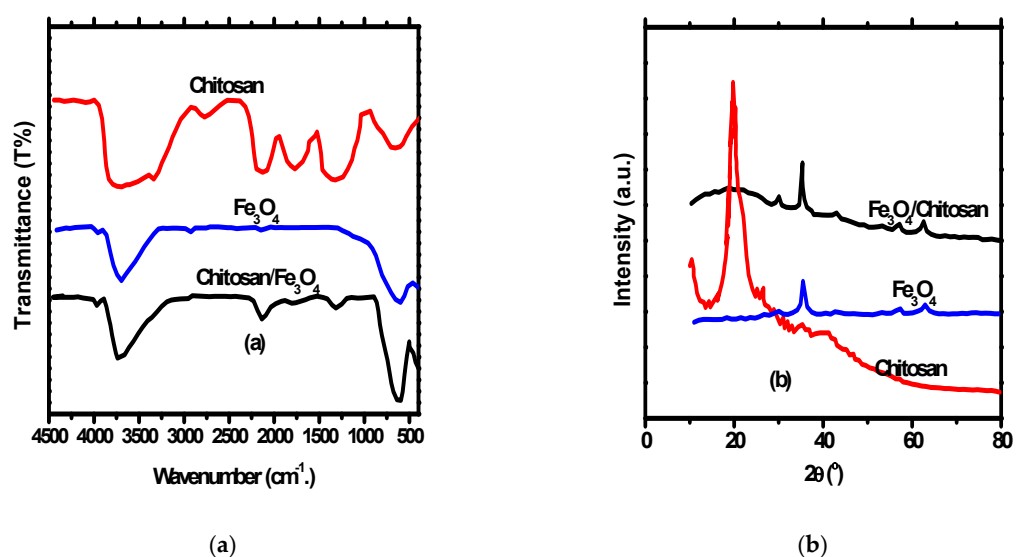

**Figure 2.** (**a**) FT-IR of chitosan, Fe₃O₄ nanoparticles and Fe₃O₄/chitosan nanocomposites; (**b**) XRD chitosan, Fe₃O₄ nanoparticles and Fe₃O₄/chitosan nanocomposites.

XRD pattern of $Fe_3O_4$ nanoparticles, chitosan, and $Fe_3O_4$/chitosan nanocomposite is shown in Figure 2b. The XRD peaks of chitosan appeared at 20.7°, while peaks appeared at 62.89°, 57.26°,

53.56°, 43.27°, 35.68°, and 30.24° for $Fe_3O_4$ nanoparticles indicating a pure spinal structure of the prepared nanoparticles. Furthermore, the appearance of the $Fe_3O_4$ nanoparticles peaks with a small shift in $Fe_3O_4$/chitosan nanocomposite XRD indicates the spinel structure of the nanocomposite. The FT-IR of nanocomposite has a band at 21.7° representing chitosan. A size of 26 nm was approved for $Fe_3O_4$ nanoparticles, and $Fe_3O_4$/chitosan nanocomposite calculated by the application of Debye–Scherrer's equation.

### 3.2. Purified MnP Measured Activity

The activity of MnP enzyme in the *A. discolor* filtrate was measured, as described in the experimental section according to the literature [38]. The MnP activity was found to be 4000 U/L, which is considered as an excellent result for the ultra-purified filtrate.

### 3.3. The Temperature and pH Effect on the Immobilized and Free MnP Activity

The immobilized MnP on the surface of $Fe_3O_4$/chitosan magnetic nanocomposite and the free MnP activities were examined at various temperatures and pHs. The working temperature ranged from 30 °C to 70 °C, while the working pH ranged from 3.5 to 9.5 as shown in Figure 3a,b, respectively. According to Figure 3a, the highest activity for both immobilized and free MnP was observed at 50 °C, with values two times higher for the immobilized enzyme (98 ± 2%). This is related to the higher stability of the immobilized enzyme compared with the free enzyme. Generally, the interaction between the support and the enzyme makes the immobilized enzyme resistant to temperature variation. The difference in activity between the immobilized and free enzymes was reported by many researchers [41,42]. According to Figure 3b, there is no significant difference between the immobilized and free MnP activity over the pH range (4.0–8.0). However, at pH = 8.5, there is a significant difference between the activity of the immobilized and the free MnP. These results might be related to the changes of enzyme conformation associated with the microenvironment change and the formation of covalent bonds upon immobilization. Consequently, the immobilization process makes the enzyme stable and resistant to pH change as reported by many researchers [43,44].

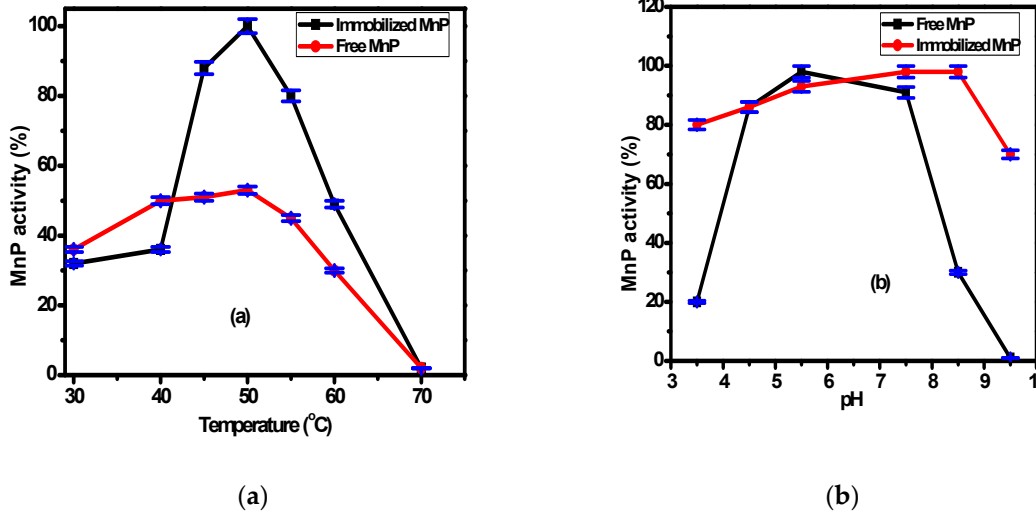

(**a**)                                                                 (**b**)

**Figure 3.** *Cont.*

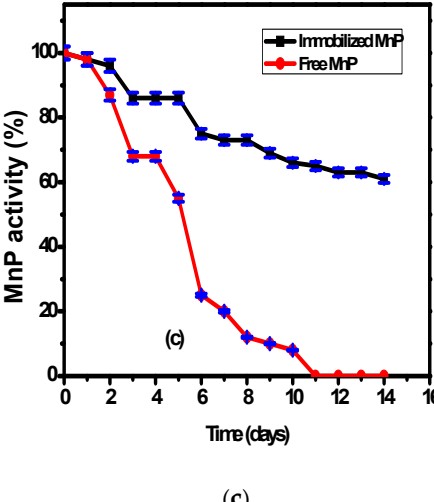

(**c**)

**Figure 3.** The effect of the operating parameters: (**a**) temperature, (**b**) pH, and (**c**) storage duration on free and immobilized MnP activity.

### 3.4. The Effect of Storage Duration on Immobilized and Free MnP Activity

The effect of storage at 20 °C on the immobilized MnP on the surface of $Fe_3O_4$/chitosan magnetic nanocomposite and on free MnP activities was studied over two weeks (Figure 3c). According to Figure 3c, there is higher stability of the immobilized MnP compared to the free enzyme during the storage period. The free enzyme showed 45 ± 2% decrease of its activity during the first five days and 90 ± 2% decrease after ten days. However, the immobilized MnP showed a slight decrease of its activity reaching 60 ± 2% after 14 days. This behavior could be related to the presence of interactions between the enzyme and the nanocomposite support, which enhance the enzymatic stability compared to the free enzyme.

### 3.5. Textile Wastewater Decolorization Using Mmobilized and Free MnP

The removal efficiencies of MB and RO 16 dyes using the $Fe_3O_4$/chitosan nanocomposite, the free and the immobilized MnP enzyme on $Fe_3O_4$/chitosan nanocomposite are shown in Figure 4a,b. Experiments were conducted at 27 °C and pH 7, the values measured for the synthetic wastewater without any adjustment. Therefore, the operating conditions (27 °C and pH 7) that mimic the real environment are more logical for the application of the process at large scales. This could reduce the process cost at large scale. According to Figure 4a,b, free MnP allowed the removal of 40 ± 2% and 43 ± 2% of the MB and RO 16 after 10 min, respectively. After 50 min, the removal efficiency of MB and RO 16 reached 52 ± 2% and 65 ± 2%, respectively. The obtained results indicate the advantages offered by the enzyme for the removal of organic pollutants from wastewater. Generally, enzymes are characterized by a notable efficiency and substrate specificity. Ligninolytic enzymes, especially MnP, can degrade organic compounds, such as polyaromatic hydrocarbons, with high efficiency [36]. They act at different environmental conditions (pH, temperature, etc.), they are effective at low pollutant concentrations and they are not inhibited by xenobiotic and microbial inhibitors [45,46]. The results suggest that the problem associated with free enzymes can be reduced by the immobilization technology [47,48].

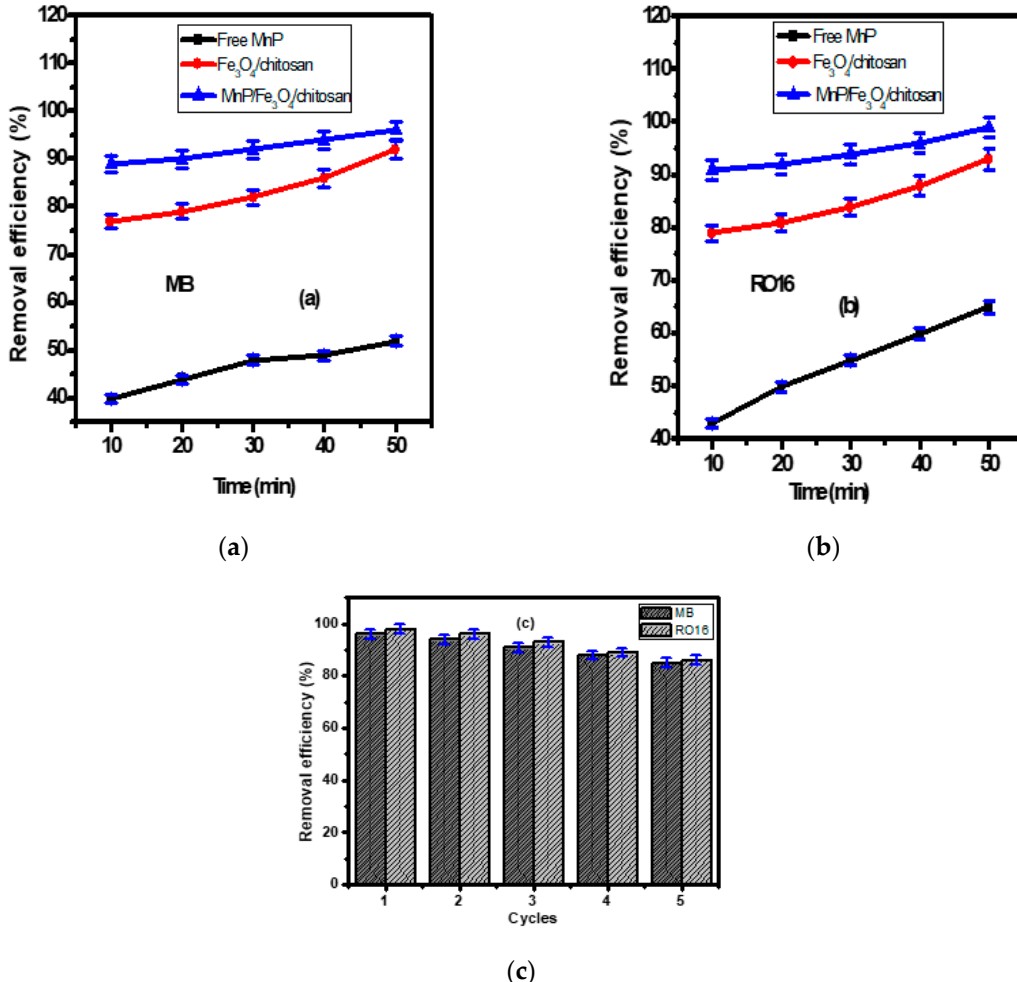

**Figure 4.** The removal of (**a**) MB and (**b**) RO 16 (dye initial concentration = 50 mg/L) by using free MnP, $Fe_3O_4$/chitosan nanocomposite, MnP/$Fe_3O_4$/chitosan material; (**c**) the reusability of MnP/$Fe_3O_4$/chitosan for five cycles.

As illustrated in Figure 4a,b, the magnetic $Fe_3O_4$/chitosan nanocomposite showed high removal efficiency for MB and RO 16 of 77 ± 2% and 79 ± 2% after 10 min, and these values reached 88 ± 2% (for MB) and 89 ± 2% (for RO 16) after 50 min, respectively. As shown in FT-IR bands (Figure 2a), the nanocomposite presents available adsorption sites, including amino and hydroxyl groups allowing an enhanced adsorption efficiency. The higher removal of both MB and RO 16 was obtained after 10 min, due to the availability of active sites on the surfaces of nanocomposite that interact easily with a large number of dye molecules. After this period, only a few active sites remain available, explaining the slight enhancement of the adsorption at the end of the experiment (50 min). This fact has been investigated by many researchers [49,50].

The magnetic $Fe_3O_4$/chitosan nanocomposite was introduced in order to immobilize the MnP enzyme. After 50 min, the immobilized enzyme allowed a decolorization rate of 96% ± 2% and 98% ± 2% for MB and RO 16, respectively (Figure 4a,b). Compared to the free enzyme, a significant improvement of the decolorization of textile wastewater was offered by the immobilized enzyme. The immobilization process increases the surface/volume ratio that enhances the binding capacity of the enzyme and helps to overcome the diffusion limitation [47]. Additionally, the small dimension of the enzyme carrier's nanoparticles can enhance the catalytic activity of the immobilized enzyme [48].

Compared to other studies, the prepared $Fe_3O_4$/chitosan nanocomposite displayed higher MB removal efficiency than other nanomaterials, such as hausmannite [51], cryptomelane ($\alpha$-$MnO_2$)

nanorods [52], and manganese oxide nanostructures [53]. Similarly, the obtained RO 16 removal efficiency was more efficient than modified zeolite [54], and humin immobilized on silica [55].

*3.6. The Reusability of MnP/Fe₃O₄/Chitosan Nanocomposite*

The reusability of $MnP/Fe_3O_4$/chitosan nanocomposite was examined due to its significant influence on the wastewater treatment processing cost, which is in part linked to the use of the nanocomposite. The nanocomposite reusability for the removal of MB and RO 16 was studied for successive five cycles as shown in Figure 4c. After each run, the magnetic nanocomposite was collected by a permanent magnet and washed many times to fit for the next run. After five cycles, the prepared $MnP/Fe_3O_4$/chitosan nanocomposite showed remarkable stability with slight decrease toward the removal of MB and RO 16 from wastewater. The removal efficiency of MB after cycle 1, 2, 3, 4, and 5 was 96 ± 2%, 94 ± 2%, 91 ± 2%, 88 ± 2%, and 85 ± 2%, respectively. Likewise, the removal efficiency of RO 16 was 98 ± 2%, 96 ± 2%, 93 ± 2%, 89 ± 2%, and 86 ± 2% after cycle 1, 2, 3, 4, and 5, respectively. These features make the prepared $MnP/Fe_3O_4$/chitosan nanocomposite a promising means of MB and RO 16 removal from wastewater.

## 4. Conclusions

$Fe_3O_4$/chitosan nanocomposite was an excellent support for the immobilization of MnP enzyme extracted from *A. discolor*. The produced material $MnP/Fe_3O_4$/chitosan was tested for decolorization of synthetic textile wastewater. Interestingly, the immobilized enzyme showed higher stability than the free enzyme at different operating conditions (pH, temperature, and storage duration). The prepared nanocomposite is considered as an ideal material with excellent properties as reported in the present study. For MB and RO 16, significant removal rates were achieved using the prepared nanocomposite. Moreover, the nanocomposite reusability was studied up to five cycles with ease recovery using a permanent magnet. However, more investigation is needed to explore other types of dyes under specific operating parameters (pH, temperature, etc.). The efficiency of this process should be studied for real textile wastewater and in real-world environmental conditions. Further, experiments using other enzymes should be conducted and the prepared nanocomposite could be explored for the bioremediation of other harmful pollutants.

**Author Contributions:** Conceptualization, F.B.R.; S.M.S., M.A.T., and W.M.; investigation, F.B.R.; data curation, F.B.R., M.A.T., and S.M.S.; writing—original draft preparation, F.B.R. and S.M.S.; writing—review and editing, W.M.; supervision, W.M.; project administration, F.B.R. and S.M.S. All authors have read and agreed to the published version of the manuscript.

**Funding:** This research was funded by Deanship of Scientific Research at King Khalid University.

**Acknowledgments:** The authors extended their appreciation to the Deanship of Scientific Research at King Khalid University for funding this work through Small Research Group Project under grant number R.G.P2/45/40.

**Conflicts of Interest:** The authors declare that there are no conflicts of interest regarding the publication of this article.

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
