# Peer review of "Iron Oxide/Chitosan Magnetic Nanocomposite Immobilized Manganese Peroxidase for Decolorization of Textile Wastewater"

_processes, doi:10.3390/pr8010005_

Round 1
Reviewer 1 Report
This Ms describes an iron oxide/chitosan magnetic nanocomposite material immobilized with the manganese peroxidase enzyme for removal of reactive orange 16 and methylene blue from textile wastewater. The spectral and surface characterizations, pH, T, stability, and reusability features are presented. Overall, the Ms appears to be suitable for publication upon addressing the below required revisions :
The title can be made shorter. Also adding the word "with" as in "immobilized with" will make it good.
In all analytical parameters measured (e.g., 96% and 98% in the abstract) , need STDEV values.
Closely relevant MNP reports for enzyme immobilization with stability and high efficiency must be cited in the Introduction (e.g., Premaratne et al., Catal. Sci. Technol. 2016, 6, 2361; Goud et al. Magnetochemistry, 2019, 5(4), 64; Nerimetla et al., Electrochimica Acta, 2018, 280, 101; Nanomaterials 2018, 8(3), 162; Nanomaterials 2019, 9(8), 1164; BioResources, 2013, 8, 2605–2619;Materials 2018, 11(8), 1312).
Typos found here and there. FOr e.g., Fig. 2b-y-axis:Intensity is spelled as "Intenesity". Also, Fig 2b-x-axis why not state the 2theta with the theta symbol instead of the text.
Fig.4-a to c----all y-axis---Efficiency is spelled wrongly.
Author Response
We wish to thank the reviewer for his comments. All the notifications included in the revised version are highlighted in red.
Comment 1: The title can be made shorter. Also adding the word "with" as in "immobilized with" will make it good.
The title has been shortened: "Iron oxide/chitosan magnetic nanocomposite immobilized manganese peroxidase for decolorization of textile wastewater" (seen Page 1, line 2-4)
Comment 2: In all analytical parameters measured (e.g., 96% and 98% in the abstract) , need STDEV values.
The SD values have been added in all the manuscript
Comment 3: Closely relevant MNP reports for enzyme immobilization with stability and high efficiency must be cited in the Introduction (e.g., Premaratne et al., Catal. Sci. Technol. 2016, 6, 2361; Goud et al. Magnetochemistry, 2019, 5(4), 64; Nerimetla et al., Electrochimica Acta, 2018, 280, 101; Nanomaterials 2018, 8(3), 162; Nanomaterials 2019, 9(8), 1164; BioResources, 2013, 8, 2605–2619;Materials 2018, 11(8), 1312).
The recommended references have been added in the introduction (references 20 to 29)
Comment 4:Typos found here and there. FOr e.g., Fig. 2b-y-axis:Intensity is spelled as "Intenesity". Also, Fig 2b-x-axis why not state the 2theta with the theta symbol instead of the text.
Fig.4-a to c--all y-axis---Efficiency is spelled wrongly.
All the manuscript has been revised and errors have been corrected. Intensity has been corrected in Fig.2b. Also, 2 theta has been changed to its symbol.
Reviewer 2 Report
1- Explain why the authors choice othe two textile dyes : reactive orange 16 and Methylene Blue?
2- In the introduction part (page 2, line 60 – 72), the authors discuss immobilization of enzymes and the usage in bioremediation as application, however, there was no citation to support the findings they relied on. Therefore, references are needed to be mentioned here.
3- Authors indicated that "Reactive orange 16 is an anionic azo dye", what about MB?
4- In Materials and Methods, full details of the companies that provided chemical should be mentioned (section 2.1). Also, provide details on the chemical characteristics of the dyes (Reactive Orange 16 and Methylene Blue) such as the dye class...
5- Page 2, line 54:" …molecules are hard to decompose in normal conditions." explain what you mean by normal conditions?
6- Page 2, line 71: " chitosan " add information related to the origin of chitosan
6- Page 6, line 198: correct “ (b)pH, and (c) storage “
7- Page 7, line 231: ….As observed in the figure, which figure?
8- Page 8 line 236: correct “MB and (b)RO 16”
9- Figure 2: Labeling on X-axis need to be clarified.
10- Section 3.5, line 212 (According to Fig. 4….). There are (a), (b) and (c) parts in Figure 4, so it should be stated clearly to which part the authors are referring.
11- Can you explain if this new process can be generalized for different classes of textile dyes?
12- In the conclusion section, can you introduce the perspective based on the obtained results.
Author Response
We wish to thank the reviewer for his comments. All the notifications included in the revised version are highlighted in red.
Comment 1- Explain why the authors choice the two textile dyes: reactive orange 16 and Methylene Blue?
A preliminary study was conducted using nanocomposite for the various dyes (Mordant red11,
Vat black 1, Vat blue 4, MB and RO 16). Interestingly, only MB and RO 16 showed efficient results. Therefore, we decide to conduct the present study for MB and RO. Now, we are planning to conduct experiments to determine the operating conditions allowing the enhancement of the removal of other dyes.
Comment 2- In the introduction part (page 2, line 60 – 72), the authors discuss immobilization of enzymes and the usage in bioremediation as application, however, there was no citation to support the findings they relied on. Therefore, references are needed to be mentioned here.
Reference 20 to 23 has been added in the text (Page 2, Line 62, 66).
Comment 3- Authors indicated that "Reactive orange 16 is an anionic azo dye", what about MB?
MB is a cationic dye. This information has been added in the text (Page 2, Line 92)
Comment 4- In Materials and Methods, full details of the companies that provided chemical should be mentioned (section 2.1). Also, provide details on the chemical characteristics of the dyes (Reactive Orange 16 and Methylene Blue) such as the dye class...
Details of companies providing chemical have been added (Page 2-3, sub-section " 2.1. Chemicals and reagents")
Reactive Orange 16 is an anionic single azo dye. Methylene Blue is a cationic thiazine dye. Information has been added in the Text (Pgag 2, line 92 and Page 2, line 94)
Comment 5- Page 2, line 54:" …molecules are hard to decompose in normal conditions." explain what you mean by normal conditions?
The word "normal" has been replaced by "natural". A statement has been added in the text "… at natural conditions (pH, temperature, etc.) and with conventional methods of treatment [9] " (Page 2, line 52).
Comment 6- Page 2, line 71: " chitosan " add information related to the origin of chitosan
Chitosan is from crab shells. This information has been added in the text, sub-section "2.1. Chemicals and reagents" (Page 2, line 90).
Comment 7- Page 6, line 198: correct “ (b)pH, and (c) storage “
Errors have been corrected (Page 6, line 205).
Comment 8- Page 7, line 231: ….As observed in the figure, which figure?
The referend figures have been added clearly for all the manuscript.
Comment 9- Page 8 line 236: correct “MB and (b)RO 16”
Errors have been corrected (Page 8, line 250)
Comment 10- Figure 2: Labeling on X-axis need to be clarified.
X axis have been clarified: in Fig.2a (wavenumber) and in Fig.2b (2θ)
Comment 11- Section 3.5, line 212 (According to Fig. 4….). There are (a), (b) and (c) parts in Figure 4, so it should be stated clearly to which part the authors are referring.
The referred figures have been stated clearly in text for the section 3.5 (Page 7, line 217-249)
Comment 12- Can you explain if this new process can be generalized for different classes of textile dyes?
A preliminary study was conducted using nanocomposite for various dyes. Interestingly, only MB and RO 16 showed efficient results. Then, we decide to conduct the present study for MB and RO 16. However, more investigations are needed to explore others dyes under specific operating conditions.
Comment 13- In the conclusion section, can you introduce the perspective based on the obtained results.
The conclusion has been slightly modified and a further research perspective has been included (Page 8/9, conclusion section, line 265)
Reviewer 3 Report
This manuscript presents many interesting data dealing mostly with the use of iron oxide/ magnetic nano composite to immobilize peroxidase enzyme which will be used for wastewater bioremediation. The Methods used are coherent since authors have performed a partial characterization of free and immobilized enzyme and its potential application for wastewater treatment. However, there are some points that could be taken into account to clarify and improve the manuscript. Here are some recommendations/suggestions towards the improvement of the manuscript
Major revision:
-Material and methods: adsorption experiment section. To evaluate the removal efficiency, authors use RO16 and MB dyes which are mixed with enzymes. Authors have not use optimal conditions of pH and temperature of free and immobilized enzymes. Based to data presented in figure 3 a, at 27°C the immobilized enzyme presented only ~30% of its full activity! Experiments at optimal conditions should therefore be performed and presented in the manuscript!
Authors should also explain the reason behind the use of conditions of pH 7 and 27°C in the adsorption experiment.
Please change also the title of this section (example: wastewater treatment).
Results and discussion. This section should be revised. there is a lack of a discussion section. Please compare the obtained results to the literature!
3.3 section: The temperature and pH effect...: the activities at 45°C and 55°C of the immobilized enzyme should be measured and added to the figure.
Lines 196-197. There is a major difference in the pH activity of free and immobilized enzymes at pH 8.5. Please, add and discuss this information!
Author Response
We wish to thank the reviewer for his comments. All the notifications included in the revised version are highlighted in red.
Comment 1:-Material and methods: adsorption experiment section. To evaluate the removal efficiency, authors use RO16 and MB dyes which are mixed with enzymes. Authors have not use optimal conditions of pH and temperature of free and immobilized enzymes. Based to data presented in figure 3 a, at 27°C the immobilized enzyme presented only ~30% of its full activity! Experiments at optimal conditions should therefore be performed and presented in the manuscript! Authors should also explain the reason behind the use of conditions of pH 7 and 27°C in the adsorption experiment.
We thank the reviewer of this important comment. In figure 3, we tried to understand the effect of pH and temperature on the enzyme activity. However, to evaluate the efficiency removal of RO 16 and MB from wastewater, experiments were conducted at 27°C and pH 7, the values measured for wastewater. Therefore, we think that is important to conduct experiments at real conductions of the wastewater without any adjustment of the temperature and the pH. Therefore, the operating conditions (27°C and pH 7) are more logical for the application of the process at large scale as it is mimic the real environment. This could reduce the process cost at large scale.
Comment 2: Please change also the title of this section (example: wastewater treatment).
The title of the sub-section "2.3 "has been changed to "Wastewater treatment experiment " (Page 4, line 143)
Comment 3: Results and discussion. This section should be revised. there is a lack of a discussion section. Please compare the obtained results to the literature!
The discussion section has been rearranged and additional comparison has been added in the sub-sections (3.1 to 3.6).
Comment 4: 3.3 section: The temperature and pH effect...: the activities at 45°C and 55°C of the immobilized enzyme should be measured and added to the figure.
The activities 45°C and 55°C of the immobilized have been measured and added into figure 3a.
Comment 5: Lines 196-197. There is a major difference in the pH activity of free and immobilized enzymes at pH 8.5. Please, add and discuss this information!
The effects of pH and temperature have been discussed (Page 6, line 191-204)
Reviewer 4 Report
The topic of this manuscript is important since textile waste is a huge problem especially in Asia where lots of textiles are produced, but also in developing countries of Africa. Textile factories need much improved waste treatments, that are effective, but also affordable.
This manuscript presents one option for textile waste water treatment. It would have been nice to have some estimations of scale up of this kind of treatments.
To make the presentation more accurate, the language should have been corrected prior to submission.
Abstract
Herein, MnP was extracted from Anthracophyllum discolor fungi and immobilized over the surface of the prepared magnetic nanocomposite Fe3O4/chitosan that save the high surface area from supporting the enzyme.
Please rewrite for better English
environmental surroundings
rather environmental factors
characterization of the synthesized biomass MnP/ Fe3O4/chitosan nanocomposite
this is no biomass?
This nanocomposite leads to the removal of 96% and 98% of MB and RO16, 31 respectively
Percentage of what kind of solution?
Introduction
l41 Uncontrolled releasing of sewage and other industrial wastes led to the introduction in the water resources of high pollutant concentrations.
rewrite for better English
l45 Due to its low prices, MB is easily acquired, so it is one of the most popular clothing colorants 45 [2].
please do not use ‘so it is’’
l49 azo dye that widely used in the dyeing process with a negative effect on the environment [6].
rather is widely used
l50 discharge of RO16 effluents inside the water resources can cause even at low concentrations many problems such as negative effect on aquatic organisms by preventing the transmission of sun beams 51 reducing the photosynthetic activity, mutagenic and carcinogenic effects
inside water resources?
So, the use of magnetic iron oxide nanoparticles as excellent support for the 63 immobilization of the enzymes provide many advantages such as the ability to quickly recover of 64 biomaterial using an external magnet, a higher amount of binding enzymes provided by magnetic 65 iron oxide nanoparticles, low operation price, less fouling, and low diffusion problem.
Language: quickly recover of biomaterial (omit of)
low diffusion problem (reduced diffusion problems)
Please do not use the word ‘so’ in scientific article
l77 Manganese peroxidase (MnP) attracts the 77 attention of researchers in the last years due to its ability
rather has attracted attention of researchers in the last years
Anthracophyllum discolor, Fe3O4/chitosan nanocomposite 83 prepared by functionalize the magnetic iron oxide nanoparticles using polymer chitosan, and MnP
please correct language ‘by functionalize the’ and rewrite .82-87 to state clearly what your study is about
Materials and Methods
MnP enzyme was extracted from A. discolor fungal strain that supplied from the microbiology 99 laboratory (biology department, Mansoura University).
please correct ‘that supplied from’ to supplied from the microbiology
Results and discussion
picture was presented in Fig.1b. According to Fig.1a and Fig.1b, there is a well distribution of Fe3O4 160 nanoparticles due to the existence of chitosan, which saves a high MnP immobilization surface.
language is bad: ‘well distribution’; ‘which saves a high MnP’
l 164the morphology of prepared nanoparticles was clear to be spherical with excellent uniformity and crystalline structure
please write in clear English what you want to say
Please give more information in fig text about
l182 Figure 1. SEM images of (a) Fe3O4 and (b) Fe3O4/chitosan; TEM images of (c) Fe3O4 and (d)
183 Fe3O4/chitosan
In fig 2a and b.
Please do not use different colors for same substance! in 2a and b.
l 196 According to Fig.3a, the higher immobilized and free MnP activity is observed at 50oC
with a clear priority to the immobilized MnP over than that of free enzyme.
rather the highest activity
‘a clear priority’ is wrong wording…
3.4. The time effect on immobilized and free MnP
Please write a better heading
Figure 4. The removal of (a) MB and (b)RO 16 by using free MnP, Fe3O4/c 236 hitosan nanocomposite,
237 MnP/Fe3O4/chitosan biomass and (c) The reusability of MnP/Fe3O4/chitosan for five cycles
please give more information about concentrations.
Put the names of substances in the figure itself
The obtained results indicate the advantages of using
215 enzymes in the adsorption of organic materials while compared to other microorganisms
Since when are organic materials microorganisms???
‘the possibility of acting in the presence of different xenobiotic materials, the
218 potential to get their substrates at very little dimensions and pores and the low response to microbial 219 inhibitors [32, 33]
this is not scientifically meaningful text
chapter 3.5 is very confusing to read, since there is no clear disposition and it contains repetitions. Please use different paragraphs to give some structure to 3.5.
Conclusions
For wastewater bioremediation, this prepared nanocomposite is considered ideal biomass
268 with excellent properties as shown from our results. For MB and RO 16 elimination, competitive
269 results of the prepared nanocomposite were achieved.
nanocomposite is not biomass! Biomass dead or alive originates from living organisms!
Author Response
We wish to thank the reviewer for his comments. All the notifications included in the revised version are highlighted in red.
Comment 1: This manuscript presents one option for textile waste water treatment. It would have been nice to have some estimations of scale up of this kind of treatments.
In this study, experiments were conducted at laboratory scale with the main objective to explore the possibility of using iron oxide/ magnetic nanocomposite to immobilize peroxidase enzyme in order to determine its efficiency for the decolorization of textile wastewater containing MB and R 16. Therefore, prior to process estimations, more investigations are needed to optimize the process and test its ability for real wastewater and at large scale. Now, we are planning other experiments in order to explore this process and to determine all the parameters affecting its suitability at large scale. This will help us to estimate the process.
Comment 2: To make the presentation more accurate, the language should have been corrected prior to submission.
The manuscript has been carefully revised and many statements have been corrected
Comment 3: Herein, MnP was extracted from Anthracophyllum discolor fungi and immobilized over the surface of the prepared magnetic nanocomposite Fe3O4/chitosan that save the high surface area from supporting the enzyme. Please rewrite for better English
The statement has been arranged (Page 1, line 20)
Comment 4 :environmental surroundings rather environmental factors
" Surroundings "has been replaced by "factors" (Page 1, line 22).
Comment 5 :
characterization of the synthesized biomass MnP/ Fe3O4/chitosan nanocomposite this is no biomass?
In all manuscript, the word "biomass" has been replaced by "material"
Comment 6:
This nanocomposite leads to the removal of 96% and 98% of MB and RO16, 31 respectively
Percentage of what kind of solution?
As indicate in the section Materials and Methods (sub section 2.5. Wastewater treatment experiment), the removal was evaluated for a synthetic wastewater containing 50 mg/L for each dye (MB and RO16).
Comment 7:Introduction
l41 Uncontrolled releasing of sewage and other industrial wastes led to the introduction in the water resources of high pollutant concentrations…….rewrite for better English
The statement has been arranged (Page 1, line 40)
Comment 8: l45 Due to its low prices, MB is easily acquired, so it is one of the most popular clothing colorants 45 [2]. please do not use ‘so it is’’
The statement has been arranged (Page 1, line 44)
Comment 9 : l49 azo dye that widely used in the dyeing process with a negative effect on the environment [6]. rather is widely used
The statement has been corrected (Page 2, line 47)
Comment 10 : l50 discharge of RO16 effluents inside the water resources can cause even at low concentrations many problems such as negative effect on aquatic organisms by preventing the transmission of sun beams 51 reducing the photosynthetic activity, mutagenic and carcinogenic effects inside water resources?
The sentence has been changed (Page 2, line 48 -50)
Comment 11: So, the use of magnetic iron oxide nanoparticles as excellent support for the 63 immobilization of the enzymes provide many advantages such as the ability to quickly recover of 64 biomaterial using an external magnet, a higher amount of binding enzymes provided by magnetic 65 iron oxide nanoparticles, low operation price, less fouling, and low diffusion problem.
Language: quickly recover of biomaterial (omit of)
low diffusion problem (reduced diffusion problems)
The paragraph has been corrected and rearranged (Page 2, line 59-66)
Comment 14 : Please do not use the word ‘so’ in scientific article
The word ‘so’ has been eliminated from the manuscript
Comment 15 : l77 Manganese peroxidase (MnP) attracts the 77 attention of researchers in the last years due to its ability
rather has attracted attention of researchers in the last years
The sentence has been corrected (Page 2, line 76-78).
Comment 16 : Anthracophyllum discolor, Fe3O4/chitosan nanocomposite 83 prepared by functionalize the magnetic iron oxide nanoparticles using polymer chitosan, and MnP
please correct language ‘by functionalize the’ and rewrite .82-87 to state clearly what your study is about
The paragraph has been corrected and rearranged (Page 2, line 81-85)
Comment 17 : Materials and Methods
MnP enzyme was extracted from A. discolor fungal strain that supplied from the microbiology 99 laboratory (biology department, Mansoura University).
please correct ‘that supplied from’ to supplied from the microbiology
The sentence has been corrected (Page 3, line 98).
Comment 18 :Results and discussion
picture was presented in Fig.1b. According to Fig.1a and Fig.1b, there is a well distribution of Fe3O4 160 nanoparticles due to the existence of chitosan, which saves a high MnP immobilization surface.
language is bad: ‘well distribution’; ‘which saves a high MnP’
The statement has been corrected (Page 4, line 159-161)
Comment 19 :l 164the morphology of prepared nanoparticles was clear to be spherical with excellent uniformity and crystalline structure please write in clear English what you want to say
The statement has been corrected (Page 4, line 163-165)
Comment 20: Please give more information in fig text about
l182 Figure 1. SEM images of (a) Fe3O4 and (b) Fe3O4/chitosan; TEM images of (c) Fe3O4 and (d)
183 Fe3O4/chitosan
In fig 2a and b.
Please do not use different colors for same substance! in 2a and b.
The figure title has been corrected. Also, the colors in the figures have been arranged as required by the reviewer.
Comment 21 : l 196 According to Fig.3a, the higher immobilized and free MnP activity is observed at 50oC
with a clear priority to the immobilized MnP over than that of free enzyme.
rather the highest activity
‘a clear priority’ is wrong wording…
The statement has been corrected (Page 6, 194-200)
Comment 22 :3.4. The time effect on immobilized and free MnP
Please write a better heading
The title of the sub-section has been arranged: "3.4. The effect of storage duration on immobilized and free MnP activity" (Page 7, line 207)
Comment 23 :Figure 4. The removal of (a) MB and (b)RO 16 by using free MnP, Fe3O4/c 236 hitosan nanocomposite,
237 MnP/Fe3O4/chitosan biomass and (c) The reusability of MnP/Fe3O4/chitosan for five cycles
please give more information about concentrations.
Put the names of substances in the figure itself
Figure titles have been arranged. Dyes initial concentrations have been added in the figure title. Names of the two dyes have been also indicated in the figure.
Comment 24 :The obtained results indicate the advantages of using
215 enzymes in the adsorption of organic materials while compared to other microorganisms
Since when are organic materials microorganisms???
The statement has been corrected (Page 7, 224-226)
Comment 25 : ‘the possibility of acting in the presence of different xenobiotic materials, the
218 potential to get their substrates at very little dimensions and pores and the low response to microbial 219 inhibitors [32, 33]
this is not scientifically meaningful text
The paragraph has been modified (Page 7, 225- 228)
Comment 26 :chapter 3.5 is very confusing to read, since there is no clear disposition and it contains repetitions. Please use different paragraphs to give some structure to 3.5.
The sub-section 3.5 has been modified (Page t, line 217)
Comment 27:Conclusions
For wastewater bioremediation, this prepared nanocomposite is considered ideal biomass
268 with excellent properties as shown from our results. For MB and RO 16 elimination, competitive
269 results of the prepared nanocomposite were achieved.
nanocomposite is not biomass! Biomass dead or alive originates from living organisms!
The conclusion has been rearranged and the word biomass has been eliminated and replaced by material. (Page 8/9, line 265-267).
Round 2
Reviewer 3 Report
Answers to all comments are convincing.
Minor point: Please add the following information in the manuscript to clarify the reason behind the use of 27C and pH 7 for wastewater remove:
To evaluate the efficiency removal of RO 16 and MB from wastewater, experiments were conducted at 27°C and pH 7, the values measured for wastewater. It is important to conduct experiments at real conductions of the wastewater without any adjustment of the temperature and the pH. Therefore, the operating conditions (27°C and pH 7) are more logical for the application of the process at large scale as it is mimic the real environment. This could reduce the process cost at large scale.
Author Response
The information has been added as recommended by the reviewer (page 7, line 220-223)
Reviewer 4 Report
The language should be improved by en expert in English. The style of writing is also not very good repeating words like 'interestingly' all through the manuscript. Thus also the style of presenting should be looked over by an experienced scientist good in English.
Author Response
The manuscript has been carefully re-revised and many statements have been corrected. All the notifications included in the new version are highlighted in red.